# A Hybrid P&O and PV Characteristics Simulation Method for GMPPT in PV Systems Under Partial Shading Conditions

**DOI:** 10.3390/s25061908

**Published:** 2025-03-19

**Authors:** Van Hien Bui, Van Du Ha, Viet Anh Truong, Thanh Long Duong

**Affiliations:** 1Faculty of Mechanical-Electrical and Computer Engineering, School of Technology, Van Lang University, Ho Chi Minh City 70000, Vietnam; 2Faculty of Electrical Engineering Technology, Industrial University of Ho Chi Minh City, Ho Chi Minh City 700000, Vietnam; 3Department of Electrical and Electronics Engineering, Thu Dau Mot University, Thu Dau Mot 820000, Vietnam; 4Faculty of Electrical and Electronics Engineering, HCMC University of Technology and Education, Ho Chi Minh City 700000, Vietnam

**Keywords:** P&O algorithm, GMPPT, partial shading conditions, open circuit voltage

## Abstract

Under uniform operating conditions, the power–voltage (P-V) and current–voltage (I-V) curves of a photovoltaic (PV) system have only one maximum point, which facilitates the operation of maximum power point tracking (MPPT) algorithms. In practice, the PV systems often operate under heterogeneous environments due to partial shading conditions (PSCs). The P-V and I-V curves exhibit multiple extremes, and distinguishing between the global maximum power point (GMPP) and local maximum power point (LMPP) is a major challenge for algorithms aiming to improve performance and convergence speed. This paper presents a global maximum power point tracking (GMPPT) method based on simulating the behavior of the I-V curve of a PV system under the influence of PSCs. With only one initial parameter selected, the proposed solution quickly determines the LMPPs based on the characteristics of the PV type and the shading condition during operation. This work helps to limit the potential GMPP region to reduce the search time and improve efficiency by using a simple algorithm and a small tuning step size. The experimental results demonstrate that the proposed method provides superior MPPT performance and significantly reduces search time due to improved GMPP detection accuracy combined with small step sizes.

## 1. Introduction

Partial shading is the cause of energy loss in photovoltaic systems during operation. Many technical solutions have been proposed to improve the efficiency of energy exploitation under this condition. They can be divided into basic groups such as the traditional solution group, the optimal or intelligence algorithm group, and the hybrid method group [1,2]. Previous research has shown that traditional methods have the advantage of being simple and easy to implement but are ineffective when partial shading conditions (PSCs) occur because they are trapped in a local maximum power point (LMPP). In contrast, the intelligent or optimized algorithm group has high performance but a complex structure, which is not economical. Combined solution groups have outstanding global maximum power point tracking (GMPPT) capabilities, but cannot avoid high costs and sometimes suffer from the pressure of calculations when having to restart in PSC cases [3,4]. Therefore, in addition to the trend of combining many algorithms in one solution to limit their disadvantages and promote their advantages, adjusting, improving or modifying traditional algorithms is also widely researched.

Today, hybrid solutions are gaining more attention and being widely applied. However, finding a solution that can fully meet the requirements such as improving performance, increasing maximum power point tracking (MPPT) speed and minimizing cost and complexity remains a challenge. Using a single decision variable, the new hybrid method that integrates the Genetic Algorithm (GA) with the Fractional Open Circuit Voltage (FOCV) method, as described in reference [5], improves both convergence speed and MPPT efficiency due to its simplified calculations. As a result, MPPT efficiency increases by 3% compared to the single FOCV solution. It streamlines the process and increases energy-harvesting efficiency, eliminating the need to disconnect the power supply to determine the reference open circuit voltage ($V_{oc}$). A drawback of this method is that the $V_{oc}$ calculation error increases due to the use of a separate sensor and its slow response. Reference [6] combined the Incremental Conductance (InC) with the Particle Swarm Optimization (PSO) algorithm to develop a two-stage approach, in which the first stage uses PSO to optimize the step size for the traditional algorithm applied in the next stage. However, the disadvantage of PSO is that the search speed and performance depend on the swarm size. In addition, the number of iterations for searching by individuals in the swarm will affect the stability of the output signals. As a result, the solution reaches a speed of 43.4 ms with an efficiency of about 99.07%. The idea of narrowing down the potential range of GMPP is also found in the paper [7] with the combination of the Improved Artificial Bee Colony (IABC) algorithm and the Simultaneous Heat-Transfer Search (SHTS) method. Their study determines the potential GMPP region based on the IABC according to the simplified probability selection mechanism. The search process is then implemented based on SHTS to find the optimal location accurately. However, like other swarm-based solutions, the convergence speed of this solution depends on the location and speed of individuals, so it is unstable and takes a long time. The advantages of Ant Colony Optimization (ACO) are combined with the accuracy and stability of Fuzzy Logic (FL) to improve MPPT efficiency in PSCs. This combination uses the ACO’s GMPPT optimization capabilities to identify the potential operating zone and then leverages FL’s efficiency around the stable operating point. The test results in reference [8] show that it can achieve 98.7% efficiency under fixed PSCs, and the search speed is about 0.88 s. However, the convergence speed of about 2.91 s is relatively slow in sudden shading change conditions.

Furthermore, enhanced versions of intelligent and optimization algorithms are continuously being developed. One of the most recent publications in this field presented a solution to automatically adjust the speed according to the objective function after each iteration for the improved Gray Wolf Optimization (GWO) algorithm [9]. As a result, the algorithm can reduce the search time by 82% to improve MPPT performance by 1.4% compared to the original version. However, the fastest convergence speed is 72 s due to the complexity of the algorithm structure. An improved variant based on traditional PSO that gradually decreases the swarm size during the search process was introduced in reference [10]. In this way, it can increase MPPT performance and reduce search time by up to 75%. However, this is not a reliable solution with a convergence speed of 0.258 s and no mention of performance. Combining the Levy Strategy with Chaos Thinking (LS-C) in an improved version of the snake optimization algorithm helps to quickly skip local search, thereby increasing search speed and improving accuracy [11]. Although this method reduced the search time by 50% compared to the original version, with a maximum efficiency of about 99.99%, the fastest speed was only 0.08 s.

Meanwhile, improvements based on traditional algorithm groups focus on reducing fluctuations around the working point and improving MPPT performance under suddenly changing operating conditions. Most studies rarely mention PSCs or must be combined with other solutions to avoid LMPP. The study in [12] introduced an improved version of the Constant Voltage (CV) method for GMPPT under PSCs. The solution uses a single parameter ($t_{max}$) that is automatically updated after each iteration to determine the optimal duty cycle (*D*) of the DC/DC converter, which is used to control the PV system operating at the GMPP. Although the solution can improve the convergence speed by up to 38.75% compared to the original version, it does not guarantee that each value of *D*, *D* = 0.2, 0.4, 0.6 and 0.8, will identify a value in an extreme region. Two measured values can fall into the same extreme region and miss untested MPPs. Therefore, the efficiency is only 99.8%. The P&O and InC algorithms mentioned in reference [13] introduced a variant with modified variable step sizes, called Modified Variable Step Size P&O (M-VSS-P&O). By checking the power deviation between two measurements to detect changes in operating conditions, the solution can achieve an MPPT efficiency of 99.37% in 16.34 ms. However, the test scenarios only focused on the uniform condition in which the I-V curve exhibits a single extremum.

Some studies belonging to the groups of traditional methods, intelligent or optimized algorithms, and combined solutions are compared and summarized in Table 1 based on aspects such as MPPT performance, convergence speed, cost, accuracy, stability, complexity and whether or not PSCs are mentioned. The results show that the traditional algorithm group is simple, easy to implement and has reasonable cost but low performance. In contrast, the remaining group of algorithms is quite complex and expensive [1]. If the ability to avoid LMPP traps is improved, the traditional algorithms group will outperform both the convergence speed and MPPT performance. Therefore, this study introduces the GMPPT solution for PV systems under PSCs based on the characteristics of the I-V curve combined with the P&O algorithm. Its outstanding performance and convergence speed are built on the following specific contributions.

Quickly determine the position of the LMPPs on the I-V curve based on a *D* value and the parameters of the photovoltaic panel.Proposed partial shading detection solution by adjusting the resistor step and the linear region on the I-V curve to narrow the MPP search range quickly.Skip the search at LMPP to accelerate convergence and improve GMPPT efficiency with small step sizes around the steady-state operating point.

The structure of the paper is as follows. Section 2 introduces the materials and methods. This section also presents the proposed solution based on a detailed analysis of the characteristics of the I-V curve. Section 3 analyzes and discusses the results obtained from the research. Finally, the conclusions are presented in Section 4.

## 2. Materials and Methods

The PV system consists of four MSX-60 [17] polycrystalline silicon solar modules, which are connected to the load through a DC/DC converter, as shown in Figure 1. The MPPT block processes the signal received from the PV system to adjust the operating position closest to the GMPP.

### 2.1. DC/DC Converter

A PV system can maximize energy extraction by adjusting the load resistance ($R_{L}$) to match the characteristic resistance at the MPP. However, directly controlling $R_{L}$ is challenging due to constantly changing operating conditions and the need to maintain stable power delivery to the load. Therefore, DC/DC converters are crucial in achieving this goal. These converters regulate the duty cycle (*D*) based on the ratio between the output voltage ($V_{out}$) and the input voltage ($V_{in}$) to adjust the equivalent resistance of the PV system, bringing it closer to the MPP. Each duty cycle value (0 < *D* < 1) corresponds to an operating point on the I-V curve. As operating conditions change, the MPP on the I-V curve shifts, requiring an adjustment in *D*. To respond quickly to rapid environmental changes, the MPPT controller employs automatic search algorithms to dynamically adjust the pulse width, allowing the system to quickly track the MPP. Among various types of DC/DC converters, the Buck-boost converter is particularly advantageous when the PV system’s output voltage varies over a wide range. It can operate in both buck and boost modes, allowing it to track GMPP throughout the PV curve [12]. The following equation gives the duty cycle of the Buck-boost converter.(1)$$\begin{array}{c} \;D=\frac{V_{out}}{V_{in}+V_{out}} \end{array}$$

### 2.2. Impact of Environment on PV Characteristics

Equation (Equation 2) represents the relationship between the current ($I_{pv}$) and voltage ($V_{pv}$) of the PV panel. The data of the MSX-60 [17] are used to investigate the I-V characteristics under different radiation and temperature conditions. The survey results in Figure 2 show that when the temperature changes, the $I_{sc}$ fluctuates insignificantly; on the contrary, it is greatly affected by the radiation changes. Furthermore, it depends linearly on both these parameters. For the $V_{oc}$, this relationship is nonlinear. Therefore, it is necessary to calculate the $V_{oc}$ value under changing radiation conditions, especially under PSCs.(2)$$\begin{array}{c} \;I_{pv}=I_{0}\left(e^{\frac{q(V_{pv}-I_{pv}R_{s})}{nkT}}-1\right)-\frac{V_{pv}+I_{pv}R_{s}}{R_{sh}} \end{array}$$
where $I_{ph}$ (A) is the light current, $I_{0}$ (A) is the diode reverse saturation current, q=1.602×$\;10^{-19}$ (C) is an electron charge, k=1.381×$\;10^{-23}$ (J/K) is the Boltzmann’s constant, T (K) is the cell temperature, n is the diode ideality factor, $R_{s}$ and $R_{sh}$ (Ω) are series and shunt resistance.

### 2.3. The Relationships at MPP of a PV Panel

Under any working condition, the $V_{oc}$ of a PV panel is determined according to (Equation 3) [18].(3)$$\begin{array}{c} V_{oc}=V_{oc\_ref}+\frac{nkT}{q}ln\frac{G}{G_{ref}} \end{array}$$
where *G* and $G_{ref}$ (W/$m^{2}$) are the radiation at operating and standard conditions, respectively. $V_{oc}$ and $V_{oc\_ref}$ (V) are the open circuit voltages at *G* and $G_{ref}$ radiation values, respectively.

Because the parameter $I_{sc}$ depends linearly on solar radiation, the $V_{oc}$ value can be determined as in (Equation 4).(4)$$\begin{array}{c} V_{oc}=V_{oc\_ref}+\frac{nkT}{q}ln\frac{I_{sc}}{I_{sc\_ref}} \end{array}$$
where $I_{sc}$ and $I_{sc\_ref}$ (A) are the short circuit current under conditions *G* and $G_{ref}$, respectively.

The above analysis indicates that $V_{oc}$ can be calculated using Equation (Equation 4) immediately after $I_{sc}$ is determined. Then, the MPP position of a PV panel is calculated based on the fill factor (FF) according to (Equation 5).(5)$$\begin{array}{c} P_{mp}=V_{mp}I_{mp}=k_{v}V_{oc}k_{i}I_{sc} \end{array}$$
where $V_{mp}$ and $I_{mp}$ are the voltage and current at the MPP; $k_{v}$ and $k_{i}$ are the proportionality constants of the voltage and current of the PV panel, respectively. By analyzing the FF factor of the PV panels studied in Figure 2, the values of $k_{v}$ and $k_{i}$ were determined to be in the range of 0.79 to 0.83 and 0.90 to 0.94, respectively. Therefore, these parameters were chosen as 0.81 and 0.92, respectively, to reduce the calculation error in this study.

Figure 3 shows the relationship between FF, $I_{sc}$, $V_{oc}$ and resistance ($R_{mp}$) at the MPP of a PV panel. This $R_{mp}$ value is determined according to (Equation 6). Meanwhile, the duty cycle of the Buck-boost converter can be rewritten from (Equation 1) as (Equation 7) below [19].(6)$$\begin{array}{c} R_{mp}=\frac{V_{mp}}{I_{mp}}=\frac{k_{v}V_{oc}}{k_{i}I_{sc}} \end{array}$$(7)$$\begin{array}{c} \frac{V_{out}}{V_{in}}=k=\frac{D}{1-D} \end{array}$$(8)$$\begin{array}{c} \Rightarrow D=\frac{k}{1+k}=1-\frac{1}{1+k} \end{array}$$
where *k* is the ratio between $V_{out}$ and $V_{in}$ of the Buck-boost converter.

The power at both sides of the DC/DC converter can be approximated by the relationship as in (Equation 9).(9)$$\begin{array}{c} \frac{V_{out}^{2}}{R_{L}}=\frac{V_{in}^{2}}{R_{pv}} \end{array}$$
where $R_{pv}$ is the resistance of the PV panel at the input of the DC/DC converter, while $R_{L}$ is the load resistance.

Combining (Equation 7) and (Equation 9), we have the relationship as in (Equation 10).(10)$$\begin{array}{c} k=\sqrt{\frac{R_{L}}{R_{pv}}} \end{array}$$

Substituting (Equation 10) into (Equation 8), we will calculate the *D* value of the PV panel according to (Equation 11), as follows.(11)$$\begin{array}{c} D=1-\frac{1}{1+\sqrt{\frac{R_{L}}{R_{pv}}}} \end{array}$$

When the PV system operates at the MPP, $R_{pv}=R_{mp}$, so $D_{mp}$ is the duty cycle value at this point, calculated according to (Equation 12), as follows.(12)$$\begin{array}{c} D_{mp}=1-\frac{1}{1+\sqrt{\frac{R_{L}}{R_{mp}}}} \end{array}$$

It seems impossible for Equation (Equation 12) to estimate a value of $D_{mp}$ from two unknown parameters, $R_{L}$ and $R_{pv}$. However, this problem is addressed using measurement results obtained from a random *D* value within the linear region of the I-V curve. This method enables the proposed algorithm to achieve two calculation objectives simultaneously. First, the current in the linear region is used to determine $I_{sc}$ based on the relationship $I_{sc}=I_{pv}$. This result can be further explained by the fact that, within the voltage range of less than 0.4$V_{oc}$, the I-V curve exhibits linear behavior [20]. To measure the parameters in this linear region, the *D* value of the Buck-boost converter must be greater than 0.69 [21]. The newly measured $I_{sc}$ value is then used to calculate the $V_{oc}$ of the PV panel according to (Equation 4). Then, $R_{mp}$, the load resistance ($R_{L}$) are determined using Equations (Equation 6) and (Equation 13), respectively. Finally, the duty cycle at the MPP ($D_{mp}$) is computed according to Equation (Equation 12).(13)$$\begin{array}{c} R_{L}=R_{pv}\frac{V_{out}^{2}}{V_{in}^{2}}=\frac{V_{pv}}{I_{pv}}\frac{V_{out}^{2}}{V_{pv}^{2}}=\frac{V_{out}^{2}}{P_{pv}} \end{array}$$

In summary, the step-by-step process to determine the MPP location of a PV panel is as follows.

1.Determine the $I_{sc}$ parameter by a value of *D* in the voltage region less than 0.4$V_{oc}$ under actual operating conditions. Reference [17] has shown that measuring the value of $I_{sc}$ in this voltage region is possible when choosing a *D* > 0.8. Measure and store the values $V_{pv}$, $I_{pv}$, $P_{pv}$, $V_{out}$ and $I_{out}$ at this *D*.2.Calculate the value of $V_{oc}$ according to $I_{sc}$ using (Equation 4).3.Estimate the value of $R_{mp}$ according to (Equation 6).4.Estimate the value of $R_{L}$ according to (Equation 13).5.Calculate the value of $D_{mp}$ using (Equation 12).

### 2.4. GMPP of the PV System Under PSCs

For PV systems working under homogeneous conditions, the characteristic curves have only one extreme, the same as the characteristic curve of a PV panel. Therefore, the approach to determining the GMPP is similar to deciding the MPP of a PV panel, as presented in Section 2.3. However, increasing the number of LMPPs will be challenging for algorithms to find the GMPP under PSCs. Earlier studies used the solution of scanning the entire I-V curve, which takes a long time and thus slows down the convergence speed. Therefore, estimating LMPPs quickly and accurately is necessary before choosing GMPP.

#### 2.4.1. Determine the MPP of the PV Module with the Most Significant Radiation

When the PV system works under PSCs, its short-circuit current ($I_{sc}$) is equal to the short-circuit current of the PV panel that receives the highest radiation $I_{sc[1]}$. It belongs to the lowest voltage region of the I-V characteristic, as shown in Figure 4. Based on this characteristic, the duty cycle at the first MPP, $D_{mp[1]}$, is determined in a similar way to that of a PV panel under actual working conditions, as in Section 2.3. It can be explained in detail as follows: under any operating condition, the $I_{sc}$ of the PV system is quickly determined based on *D* = 0.8 and is always equal to $I_{sc[1]}$ [17]. Next, using (Equation 4), we will calculate the value of $V_{oc[1]}$. Then, the parameters of the first region on the I-V characteristic, including $P_{mp[1]},R_{mp[1]},R_{L}$ and $D_{mp[1]}$, are determined based on (Equation 5), (Equation 6), (Equation 13) and (Equation 12), respectively. In other words, these parameters are always determined based on $I_{sc}$, regardless of any changes in the I-V curve. They are used as reference values to detect partial shading in the following steps.

#### 2.4.2. Determine the Rest of the MPPs in the PV System

Assuming the PV system operates under uniform conditions, the value $V_{oc[2]}=2V_{oc[1]}$, so at the MPP of the second step region ($M_{2}$), the resistance value ($R_{mp[2]}$) is estimated according to (Equation 14). In this case, the line representing $R_{mp[2]}$ intersects the I-V curve at point A, which coincides with $M_{2}$. However, suppose the PV system operates under inhomogeneous conditions. In that case, the MPP of the second step region will shift to $M_{2^{\prime }}$ or $M_{2\prime \prime }$, depending on the shading level. So, the intersection point between the line representing $R_{mp[2]}$ and the I-V curve will be $A^{\prime }$ or $A^{\prime \prime }$, which does not coincide with $M_{2^{\prime }}$ or $A_{2\prime \prime }$ (Figure 4).(14)$$\begin{array}{c} R_{mp[2]}=R_{mp[1]}+\frac{k_{v}V_{oc[1]}}{k_{i}I_{sc[1]}} \end{array}$$

The value of $D_{mp[2]}$ will be calculated according to $R_{mp[2]}$ using Equation (Equation 12). Measure the operating parameters corresponding to $D_{mp[2]}$ value to determine the PV system’s operating voltage, $\left(V_{pv}\right)$. If the operating conditions are uniform, the system will work at point A, where its voltage is $V_{A}=V_{mp[2]}$, which is also the second MPP location, $M_{2}$. In contrast, if shading appears, the voltage value received at $A^{\prime }$ or $A^{\prime \prime }$ depends on the level of shading, as shown in Figure 4. The solution will check the shading phenomenon by comparing $V_{pv}$ with $1.4V_{oc[1]}$. This parameter can be explained as follows: Our previous study has shown that in the voltage range less than $0.4V_{oc}$, the I-V curve of each PV panel is linear [20]. Therefore, when considering the possibility of shading in the second region, the testing position must be moved by a $V_{oc[1]}$ value. The intersection between the line containing $R_{mp[2]}$ and the I-V curve relative to the position $1.4V_{oc[1]}$ will change according to the shading condition. Therefore, the principle of checking the partial shading condition is proposed as follows.

If $V_{pv}$ < 1.4$V_{oc[1]}$, this means that the PV system operates under PSCs, and $I_{sc[2]}$ decreases sharply compared to $I_{sc[1]}$. The measured voltage, in this case, has a value in the range from $V_{mp[1]}$ to 1.4$V_{oc[1]}$ (at $A^{\prime \prime }$). If solar radiation drops sharply, the position of point $A^{\prime \prime }$ approaches $M_{1}$, and choosing the value $I_{sc[2]}$ = $I_{A\prime \prime }$ will give less accurate results. To pass the nonlinear region (from $M_{1}$ to $A^{\prime \prime }$), reference [22] adjusted the step size (ΔD) and checked the current difference between the two measurements until this value was less than $dI_{min}$. This work ensures that the working point has been searched in the next MPP region and can pass the LMPP. However, the proposed solution faces three problems. First, scanning the entire I-V curve to check the shading conditions is necessary, so it takes more time to search. Second, using the same ΔD value to test the shading condition and search for GMPP will make it difficult to simultaneously achieve the two goals of improving performance and convergence speed. This drawback can be explained in detail by stating that, with a small ΔD value chosen to improve performance, many adjustment steps are needed to jump over the nonlinear region due to shading. Therefore, the convergence speed of the entire solution will be significantly reduced. In contrast, when using a large ΔD for fast positioning of LMPPs, the performance decreases and becomes less stable around the working point. Third, to find the first MPP location, the solution of the 0.8$V_{oc}$ location reference and searching using the InC algorithm takes more time. This drawback is addressed in reference [23] by employing a significantly large initial ΔD value, allowing for a faster convergence to the MPP. Then, variable steps are used to improve performance. However, comparing the voltage at the current GMPP with the voltage value at other points of the same power to check the PSC causes the solution to need two checks for each new MPP peak. Therefore, the convergence speed is relatively low. Recently, reference [24] proposed a strategy to spread the test locations evenly within the voltage limit of the entire PV system. This solution is quite simple and effective when the number of PVs in the string is small and the radiation is evenly reduced on the panels. However, when solar radiation changes are not uniform, uncertainty exists regarding whether all selected points are close to the LMPP and whether the two test locations may fall within the same MPP region.

This study overcomes the shortcomings of previous research using the 0.4$V_{oc}$ position for PSC testing. The justification for using the proposed 0.4$V_{oc}$ point in PSC verification is as follows: First, when irradiance decreases, the line containing $R_{mp[2]}$ can intersect the I-V characteristic at either $A^{\prime }$ or $A^{\prime \prime }$, depending on the percentage of shading (Figure 4). If the system operates under uniform conditions or with light partial shading, the measured voltages at $A^{\prime }$ or $A^{\prime \prime }$ are greater than $V_{oc[1]}$ + 0.4$V_{oc[1]}$. Consequently, the current values at these points are selected as short-circuit currents in the second region ($I_{sc[2]}$). These currents are also higher than those at the MPP in the second region (at $M_{2^{\prime }}$ or $M_{2\prime \prime }$), ensuring that the MPP is located within this region without shifting to the third region. Second, within the 0.4$V_{oc}$ limit, the I-V characteristic remains linear, making the determination of $I_{sc}$ in this region more accurate. However, under significantly reduced irradiance (represented by the red characteristic line), the operating point $A^{\prime \prime }$ may retreat into the nonlinear region (from $M_{1}$ to $V_{oc[1]}$). If the measured values in this region are used to determine the MPP, they remain in Region 1. In these cases, shifting the operating point to Region 2 is necessary by increasing the resistance value $R_{mp[2]}$. Therefore, the 0.4$V_{oc}$ threshold is an optimal limit for updating the operating region, ensuring that no step region is overlooked. To transition to the second step region, the $R_{mp[2]}$ value should be increased according to (Equation 15), and the $D_{mp[2]}$ value should be recalculated using Equation (Equation 12) until $V_{A\prime \prime }>1.4V_{oc[1]}$.(15)$$\begin{array}{c} R_{mp[2]}=R_{mp[2]}+\lambda \frac{V_{oc[1]}}{I_{sc[1]}} \end{array}$$

Adjusting ΔR to rapidly transition to the next MPP region will reduce the algorithm’s iteration count because a small *D* can be used for searching after the potential GMPP region has been identified. This results in an increase in MPPT efficiency. This operating principle is the same as choosing an enormous ΔD value at startup and reducing the step size when approaching the MPP. However, adjusting based on ΔR in an early stage will ensure an accurate estimate of the distance to the reference location (0.4$V_{oc}$), where the measured current value is still in the linear region on the I-V curve and has minimal error relative to $I_{sc}$. This advantage has been demonstrated in our previous study [21].

If $V_{pv}>1.4V_{oc[1]}$, then the PV system is probably working under uniform conditions (at point A) or slightly reduced radiation (at point $A^{\prime }$). Therefore, the operating current ($I_{pv}$) at this location is chosen as $I_{sc[2]}$ to calculate $V_{oc[2]}$ according to (Equation 16).(16)$$\begin{array}{c} V_{oc[2]}=V_{oc[1]}+V_{oc\_ref}+\frac{nkT}{q}ln\frac{I_{sc[2]}}{I_{sc\_ref}} \end{array}$$

Calculate the values of $P_{mp[2]},R_{mp[2]}$ and $D_{mp[2]}$ based on the two parameters $I_{sc[2]}$ and $V_{oc[2]}$, then measure and update the parameters at $D_{mp[2]}$. Check the remaining panels in the PV system similarly and store all MPP locations. The most significant power value among the LMPPs is selected as the potential GMPP location for implementing the P&O algorithm to find the exact value of the GMPP.

### 2.5. Proposed GMPPT Method

Based on the analysis above, to determine the GMPP of the PV system under PSCs, the proposed algorithm is as follows:1.Step 1. Set the values *D* = 0.8 and the step size ΔD = 0.015 to measure $I_{sc}$ of the system, which consists of N panels in the PV system. Select the value $I_{sc[1]}=I_{sc}$ is the short-circuit current of the first region on the I-V curve. Measure and store values at *D*, including $V_{pv}$, $I_{pv}$, $P_{pv}$, $V_{out}$ and $I_{out}$.2.Step 2. Calculate $V_{oc[1]}$ of a first region based on (Equation 4).3.Step 3. Estimating the parameters at the first MPP point, including $P_{mp[1]}$, $R_{mp[1]},R_{L}$ and $D_{mp[1]}$, according to (Equation 5), (Equation 6), (Equation 13) and (Equation 12), respectively. Then update the measured values at $D_{mp[1]}$.4.Step 4. Determine the working resistance at the $i^{th}$ MPP points (i = 2 to N) according to (Equation 17).(17)$$\begin{array}{c} R_{mp[i]}=R_{mp[i-1]}+\frac{k_{v}V_{oc[1]}}{k_{i}I_{sc[1]}} \end{array}$$5.Step 5. Calculate $D_{mp[i]}$ values according to (Equation 18).(18)$$\begin{array}{c} D_{mp[i]}=1-\frac{1}{1+\sqrt{\frac{R_{L}}{R_{mp[i]}}}} \end{array}$$6.Step 6. Measure and store the parameters $I_{pv[i]},V_{pv[i]}$ and $P_{pv[i]}$ corresponding to the $D_{mp[i]}$ value.7.Step 7. Check the working position in shaded areas.If $V_{pv[i]}<V_{oc[i-1]}+0.4V_{oc[1]}$, then $I_{vp[i]}\neq I_{sc[i]}$, so the controller must increase the $R_{mp[i]}$ value according to (Equation 19) and return to step 5. Otherwise, go to the next step.(19)$$\begin{array}{c} R_{mp[i]}=R_{mp[i]}+\lambda \frac{k_{v}V_{oc[1]}}{k_{i}I_{sc[1]}} \end{array}$$If $V_{pv[i]}>V_{oc[i-1]}+0.4V_{oc[1]}$ then $I_{sc[i]}=I_{pv[i]}$. Calculate the $V_{oc[i]}$ value at the i^*th*^ MPP according to (Equation 20).(20)$$\begin{array}{c} V_{oc[i]}=V_{oc[i-1]}+V_{oc\_ref}+\frac{nkT}{q}ln\frac{I_{sc[i]}}{I_{sc\_ref}} \end{array}$$Calculate the current and voltage values at the i^*th*^ MPP according to (Equation 21) and (Equation 22).(21)$$\begin{array}{c} I_{mp[i]}=k_{i}I_{sc[i]}=k_{i}I_{pv[i]} \end{array}$$(22)$$\begin{array}{c} V_{mp[i]}=k_{v}V_{oc[i]} \end{array}$$Determine the $P_{mp[i]}$ value according to (Equation 23) and recalculate $R_{mp[i]}$ following (Equation 24).(23)$$\begin{array}{c} P_{mp[i]}=V_{mp[i]}I_{mp[i]}=k_{v}V_{oc[i]}k_{i}I_{sc[i]} \end{array}$$(24)$$\begin{array}{c} R_{mp[i]}=\frac{V_{mp[i]}}{I_{mp[i]}} \end{array}$$8.Step 8. Calculate and store $P_{pv[i]}$ values according to $D_{mp[i]}$ to find the maximum power and corresponding *D* value.9.Step 9. Select the reference power, $P_{ref}$, corresponding to the largest $P_{pv[i]}$ value of the previous step. At the same time, select the values $D_{ref}$, $V_{ref}$ and $I_{ref}$ corresponding to $P_{ref}$ as reference values for the next steps.10.Step 10. Check the convergence condition.Modify the parameter ΔD to monitor the resulting power and voltage outputs and then compare them with their corresponding values in the previous step to determine the optimal MPP. The algorithm reaches convergence by satisfying the constraint outlined in (Equation 25).(25)$$\begin{array}{c} \left|\Delta P\right|=\frac{P_{i+1}-P_{i}}{P_{i}}100\%\leq \varepsilon_{p} \end{array}$$If the condition in (Equation 25) is not satisfied, the voltage error ΔV is checked according to (Equation 26).(26)$$\begin{array}{c} \left|\Delta V\right|=\frac{V_{i+1}-V_{i}}{V_{i}}100\%\leq \varepsilon_{v} \end{array}$$If ΔVΔP>0, reduce *D* to increase V; otherwise, increase *D* to reduce V.If the current experiences a significant deviation due to changes in operating conditions, the solution will restart the calculation from the initial MPP. Figure 5 shows the flow chart of the proposed algorithm.(27)$$\begin{array}{c} \left|\Delta I\right|=\frac{I_{i+1}-I_{i}}{I_{i}}100\%\leq \varepsilon_{i} \end{array}$$

## 3. Results and Discussion

The GMPPT algorithm is validated using simulation in the PSIM environment (version 9.1.3) and experiments with the Chroma Simulator (version CHROMA ATE,62050H-600S,00104,02.10). Table 2 presents the proposed study cases, in which the location of the GMPP is randomly distributed among the LMPPs within the entire operating voltage range. Details on locations and graphs depicting the PV system behavior for these cases are presented in Figure 6. The simulation results focus on highlighting the values of the proposed solution, included.

1.Validate the ability to accurately position LMPPs on the PV system’s I-V characteristic curve under PSCs based on an initial reference position.2.Confirming the effectiveness and simplicity of the proposed solution in improving MPPT performance and speed under PSCs.3.Demonstrate the superior LMPP trap avoidance ability of the proposed method by simulating the I-V curve under various operating conditions.

When the irradiance changes, the $I_{sc}$ and $V_{oc}$ parameters of the PV system fluctuate but have little impact on determining the location of the first MPP. This property results from $I_{sc}$ being directly proportional to solar irradiance. If $I_{sc}$ is accurately determined, $V_{oc}$ will be precisely calculated using (Equation 4). As analyzed earlier, with the initial duty cycle *D* selected in this study, the measurement of $I_{sc}$ is consistently ensured to be within the linear region. Consequently, the error in calculating $V_{oc}$ and the MPP position is negligible. The I-V characteristics in Figure 6 also indicate that the voltage $V_{oc[1]}$ remains relatively stable under all test conditions, and the voltage at the first MPP position, $V_{mp[1]}$, is nearly constant. This result facilitates the accurate determination of $D_{mp[1]}$ and serves as a reference value for implementing a partial shading detection algorithm on the I-V curve. However, varying irradiance conditions influence the speed of shading detection in the PV system. This phenomenon can be explained using Figure 4: Under a small decrease in irradiance, a slight adjustment of the resistance value is enough to move from the intersection point $A^{\prime }$ to position $M_{2^{\prime }}$. In contrast, a significant decrease in irradiance, moving from $A^{\prime \prime }$ to $M_{2\prime \prime }$ requires multiple resistance updates. Nonetheless, resistance adjustments are performed independently of the duty cycle search loops. Therefore, the proposed method ensures stability output waveform, as adjustments to *D* are only performed around the GMPP. The following simulation results and experimental results will validate this approach.

### 3.1. Simulation Results

When simulating under testing conditions, the proposed algorithm finds the GMPP zone in less than 12 ms. Therefore, the time needed to determine the stable working point of the PV system is fast. The output waveforms can be explained based on the simulation graphs in Figure 7 and Figure 8 as follows.

Under uniform operating conditions, from the initial value, *D* = 0.80, the $D_{mp[i]}$ values are calculated with a total time of 11.67 ms. Therefore, the P&O algorithm only needs four adjustment steps to converge at $D_{mp}$ with 18.93 ms. The maximum estimated power is $P_{mp[4]}$ = 144.94 W at $D_{mp[4]}$, which is also the operating value of the PV system at the convergence point $D_{mp}$ in this case, with an MPPT efficiency of 99.93% (Figure 7).

Under partial shade conditions, in Figure 8, the P-V waveform shows that the GMPP value at $P_{mp[1]}$ is approximately $P_{mp[4]}$. Algorithms with poor accuracy are often trapped in LMPP in similar cases. However, with the ability to accurately estimate and adjust appropriate parameters, the proposed solution has excelled in determining the correct GMPP. This advantage can be explained by the duty cycle waveform, with the initial value *D* = 0.8 used to determine the $I_{sc}$ of the PV system. Then, the $D_{mp[i]}$ and $P_{mp[i]}$ values are calculated in 11.7 ms. The solution selects the value $D_{mp[1]}$ = 0.6057 as a reference point to check the optimal working position from this data. Through 6 adjustment steps, the P&O algorithm converges at $D_{mp}$ = 0.5925 after that. The total time for the proposed solution to converge at GMPP is 25.71 ms. The output power stabilized at the optimal position immediately after that, with about 58.05 W reaching 100%.

The simulation results in Table 3 show that the reference $D_{ref}$ value has an error from the convergence position $D_{mp}$ of about 0.015 to 0.030. With the step size ΔD = 0.015 set for this proposed algorithm, the convergence point is far from the reference position about two adjustment steps. This result contributes to increasing the search speed of the proposed solution. This advantage is demonstrated in the power deviation value (ΔP%) between the estimated power at the reference position ($P_{ref}$) and the power obtained at the position of GMPP ($P_{mp}$) in Table 3. Under uniform conditions, this parameter has a minimum value of only 0.14%, while the maximum error is 6.98% under PSC conditions. Although the survey cases exhibit different characteristic curves, number of LMPPs and GMPP positions, their convergence speeds are nearly identical. This result can be explained as follows: the computation time for determining the $D_{mp[i]}$ positions is almost constant across different operating scenarios. As a result, the convergence speed is primarily influenced by the accuracy of the estimated GMPP position. If this position is accurately predicted, only a few iterations are required for convergence. The ability of the proposed solution to estimate LMPP values with high precision requires a nearly consistent number of adjustment steps to achieve convergence at the optimal GMPP. This result shows that in the calculation results without the P&O algorithm, the estimated value at the GMPPs can be over 93% with an average error of less than 4%. However, several iterations can be performed through the traditional P&O algorithm to ensure maximum power is obtained. The fewer iterations the algorithm uses, the more stable the output waveform is, thus improving the convergence speed and MPPT efficiency. Because each time *D* is adjusted, the output waveform will be affected, making it less stable. The average MPPT efficiency in the simulated cases is over 99.97%, with the fastest search speed of about 18.93 ms. It shows superior LMPP trapping avoidance ability and accurately predicts the GMPP location of the PV system under PSCs.

Another scenario proposed for simulation under continuously varying irradiance conditions. The PV system works under uniform conditions and the PSC occurs at 0.1 s. The radiation on PV panels varies, and the GMPP belongs to the first region of the four LMPPs on the I-V characteristic curve, Figure 9. The proposed solution demonstrated the ability to track the new GMPP as it rapidly moves to the alternate working point and stabilizes at 0.13 s. After that, at 0.2 s, the irradiance continued to change, resulting in the PV system generating four LMPPs. The GMPP location belongs to the third region on the I-V curve. The algorithm further demonstrated its ability to effectively adjust the working position when it stabilized at 0.225 s. When the uniform working condition was restored at 0.3 s, it responded quickly and efficiently, achieving an average convergence speed of 0.022 s and an MPPT efficiency of over 99% under continuously changing operating conditions. The results showed that the algorithm could respond flexibly under uniform conditions, PSCs and constantly changing operating conditions.

### 3.2. Experimental Results

The proposed algorithm is tested using a PV Chroma 62050H-600S simulator for the cases in Table 2. This study employs a Buck-boost converter to control the PV system operating at the GMPP. The main parameters of the DC/DC converter include an input capacitance ($C_{in}$ = 100 μF), an output capacitance ($C_{out}$ = 47 μF), an inductance (L = 0.4 mH) and a load resistance of $R_{L}$ = 20 Ω, the switching frequency is 50 kHz. In this study, an Arduino Nano ATmega328P controls the DC/DC converter, providing an optimal *D* value for the PV system operating at the GMPP. The experimental setup, as shown in Figure 10, is used to analyze the PV system’s output waveforms under different conditions. A summary of the output power and experimental performance is presented in Table 4, showing an average GMPPT efficiency of approximately 99.48%, with a maximum extraction of 99.80%. The specific experimental scenarios are as follows.

When operating under uniform conditions (No.1), the output power is approximately 99.75% the maximum power of the PV system (Figure 11). The results indicate that the short circuit current value of the PV system, determined from the initial position *D* = 0.8, is also the short circuit current in the first step region ($I_{sc[1]}$). Then, the LMPP values in the regions are determined and updated sequentially. The current waveform shows that the $I_{sc[i]}$ values in different regions do not differ significantly. However, the current in zone 4 is slightly lower than in the other regions because the measurement point falls within the transition region between the linear and nonlinear segments of the I-V curve.

Meanwhile, the voltage and power waveforms continue to increase steadily because the solution searches from low-voltage to high-voltage regions, and only one extremum exists on the P-V curve. The potential GMPP limit belongs to Zone 4, and the proposed solution performs a maximum power value check there based on the P&O method. Several adjustment steps are required to ensure the maximum possible output power extraction. Since there is no significant current drop between regions, the calculation points do not need to update the increment ΔR to adjust the operating region.

When operating under PSC (No.2), the principle of determining $I_{sc}$ and the MPP in the first step region is the same as under uniform conditions. However, the current tends to decrease in the subsequent regions (Figure 12). The current in Zone 2 is lower than in Zone 1, but the maximum power still increases due to the increasing voltage level. The same phenomenon occurs in Region 3. However, when transitioning from Region 2 to Region 3, the solution must adjust the resistor to ensure that the two measured maximum values do not belong to the same step region (i.e., they do not have the same current value). This operating mechanism repeats when switching the search region from 3 to 4. At the end of the PSC checking, the MPP in Region 3 is the highest compared to the other regions. Therefore, the algorithm confines the potential GMPP to this region to optimize the search for the GMPP value. The current waveform in Figure 12b shows that after scanning for PSCs across the entire I-V curve, the current reverses from the lowest region, Region 4, back to Region 3. Meanwhile, the voltage in Region 3 is lower than in Region 4, causing the output voltage waveform to drop suddenly. As a result, the power reaches its maximum efficiency of approximately 99.80%.

To compare the effectiveness of the proposed solution with the traditional version, both versions use the same implementation principle of the P&O algorithm. They are initialized with a starting value of *D* = 0.8 and a step size of ΔD = 0.015. The key difference is that the traditional P&O algorithm searches directly from the starting position until it finds the nearest extreme value. Meanwhile, the proposed solution performs independent calculations to determine the LMPPs and identify the potential GMPP limit in the first stage. After narrowing the search area, the P&O algorithm is then deployed to determine the convergence point.

The experimentally obtained output waveforms demonstrate that the proposed solution is more effective than the traditional P&O method in the following aspects: First, it has a superior ability to avoid LMPP traps by simulating I-V characteristics under different operating conditions. The potential GMPP region is identified among the LMPPs to ensure that no extreme points on the I-V curve are overlooked. While, the traditional P&O method always identifies the first extremum in the low voltage region when starting from the initial value, *D* = 0.8, (Figure 13). This principle indicates that the traditional P&O method can correctly identify only two cases: No.1 (where there is a single extremum) and No.5 (where the extremum belongs to Region 1). Therefore, the experimental results for Case 2 are trapped in LMPP, resulting in an efficiency only slightly above 56.37%.

Second, the output signal of the proposed method is more stable than that of the traditional P&O method. In the old version, the output waveform is affected by continuous fluctuations due to changes in step size ΔD. Therefore, the convergence speed and performance of the traditional P&O method depend on this parameter. A small ΔD will improve the performance of the PV system, but numerous adjustments are required, causing continuous fluctuations in the output waveform and a slow convergence speed. In contrast, if ΔD is large, fewer iterations are needed, resulting in faster convergence; however, efficiency decreases, and the output signal at the operating point becomes unstable. This drawback can be observed in the output waveform shown in Figure 14. Although it achieves MPPT efficiency equivalent to that of the proposed solution due to the presence of only one extremum in the I-V characteristic, its output waveform converges 1 s slower. The proposed solution mitigates this drawback by using a significant adjustment of ΔR to quickly change the search area. The value of *D* remains stable while updating the new position based on *R*, minimizing the number of iterations. Therefore, the proposed solution can utilize a small ΔD value to enhance GMPPT performance and stabilize the output signal without compromising the convergence speed. The output power waveform in Figure 11 shows this feature. Although there is only one extremum, the solution introduces four resting periods to reposition the search area, whereas the traditional P&O method continuously loops to reach the MPP.

The results of applying the proposed solution to GMPPT for PV systems operating under PSCs are used to compare with previously published research results listed in Table 5, showing that traditional solutions often fail to validate the GMPPT capability under PSCs. In this group, the M-VSS-P&O algorithm [13] has the fastest speed of 0.016 s, while the CC solution [16] has the most significant efficiency of 99.88%. Although the CV method [1] can achieve 100% MPPT efficiency, the convergence speed is not mentioned. In contrast, the group of optimal or hybrid solutions with improvements allows them to perform better under PSCs. In addition to the proposed solution, all algorithms can achieve one single goal: to improve GMPPT efficiency or convergence speed. With a time of 0.018 s and outstanding performance, the proposed solution has great potential for exploitation and application in series PV systems operating under PSCs.

The proposed solution follows the same operating principle as the conventional P&O algorithm, which has been demonstrated to be less complex than other algorithms [1,2]. In addition, it reduces costs compared to the unimproved version, as it does not require separate voltage and current sensors. Instead, these parameters are computed directly from a single initial *D* under specific operating conditions. As a result, the proposed method not only improves performance and convergence speed under PSCs but is also simple, easy to implement and highly accurate. The simulation and experimental results show that the proposed algorithm has outstanding GMPP efficiency in all situations.

## 4. Conclusions

This paper presents a GMPPT solution for a series-connected PV system operating under PSCs and continuously changing operating conditions. By quickly calculating the $I_{sc}$ and $V_{oc}$ values of the PV panel receiving the maximum irradiance in the PV system based on a starting value and the characteristics of the PV type, the proposed solution locates the first MPP on the I-V curve more quickly and accurately. Then, the shading condition is determined based on the $R_{pv}$ adjustment step and the $0.4V_{oc}$ reference position on each shading area without having to scan the entire I-V curve. This advantage helps to quickly limit the search space, and reduce the number of iterations, so the output signal is more stable than previous solutions. Adjusting the search area based on $R_{pv}$ is also faster than using the ΔD step size value because the algorithm needs to improve the GMPPT performance with a small ΔD value. The results show that this method is simpler, easier to implement and more reliable than previously proposed algorithms under PSCs. The simulation efficiency can reach 100%, while the experimental efficiency is 99.80%. The convergence speed is also faster than the recently published optimization algorithms. These advantages show that it has many potential applications in PV systems operating under different conditions. However, its disadvantage is that it depends on the type of PV. Each PV type has various parameters for calculating $V_{oc}$. This technical gap is also the problem that the research team needs to complete in future work to have a standard formula for determining the first MPP position on the I-V curve when PSCs occur.

## Figures and Tables

**Figure 1 sensors-25-01908-f001:**
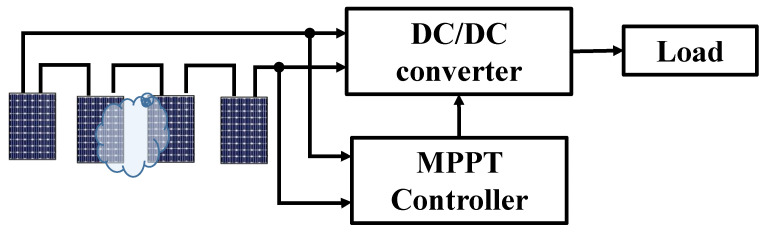
Block diagram of the proposed solution.

**Figure 2 sensors-25-01908-f002:**
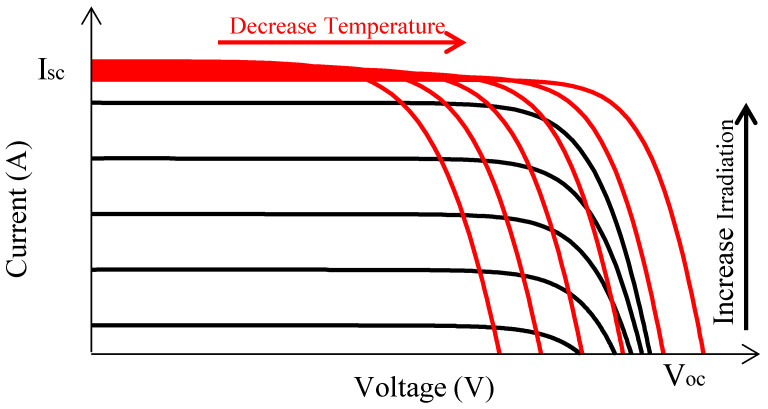
I-V characteristics of a PV panel under different conditions.

**Figure 3 sensors-25-01908-f003:**
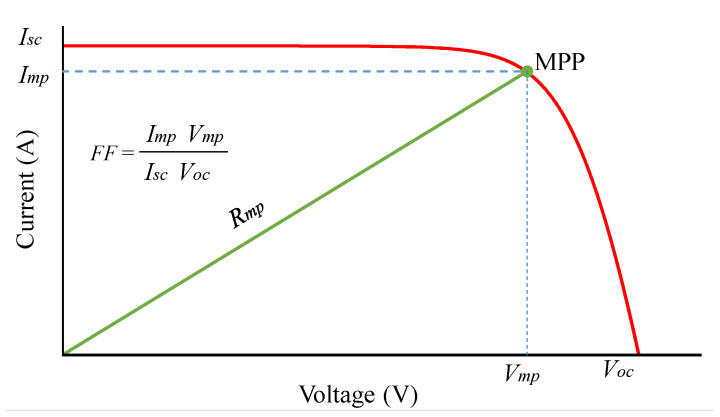
The fill factor of the PV module.

**Figure 4 sensors-25-01908-f004:**
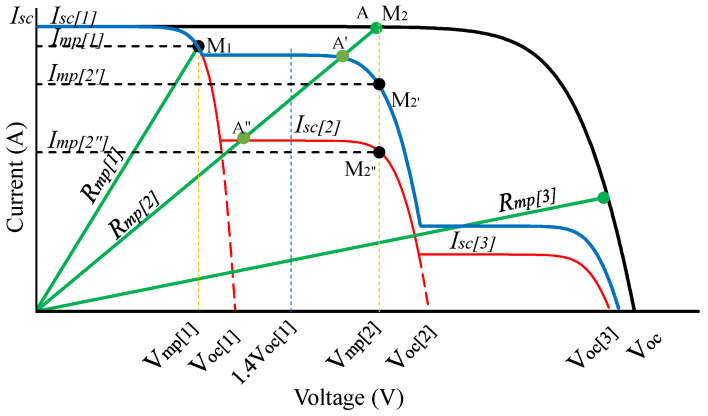
Principle of checking shading condition on PV system; black line: I-V curve under uniform conditions; blue line: I-V curve under slightly reduced irradiance; red line: I-V curve under significantly reduced irradiance; green line: line representing $R_{mp[i]}$. dashed black line: current at MPPs; dashed yellow line: voltage at MPPs. dashed red line: curve passing through $V_{oc[i]}$; dashed blue line: line representing the limit at 1.4$V_{oc[1]}$.

**Figure 5 sensors-25-01908-f005:**
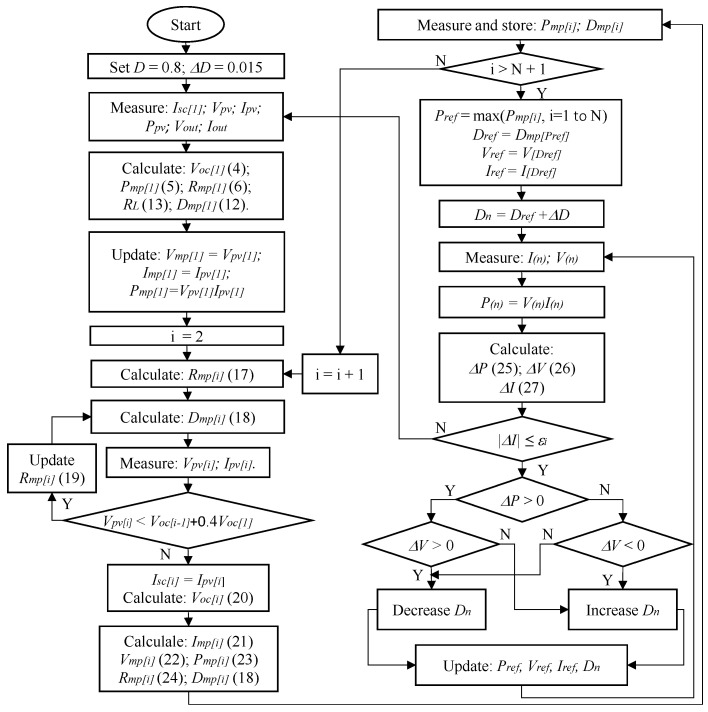
Flow chart of the proposed algorithm.

**Figure 6 sensors-25-01908-f006:**
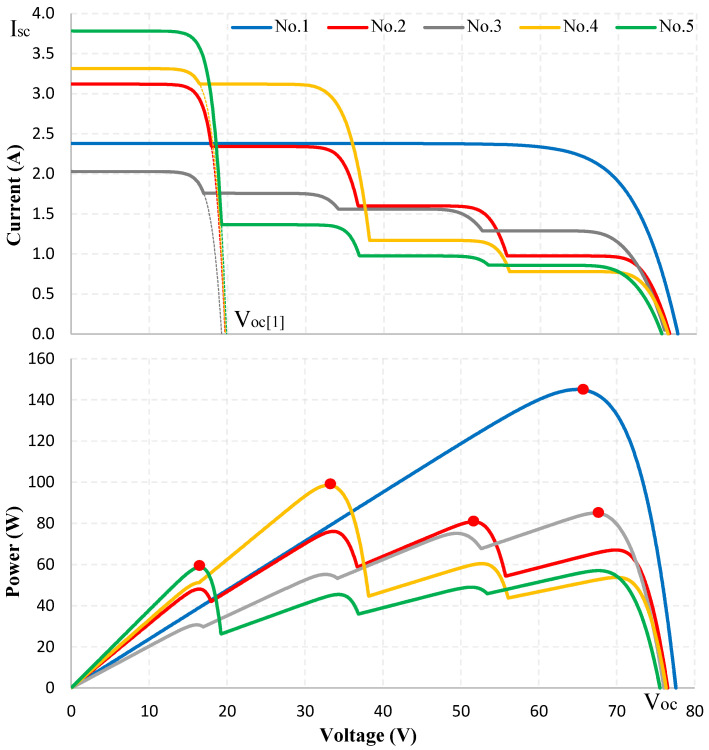
Positions of MPPs on characteristic curves (red dots) under PSC.

**Figure 7 sensors-25-01908-f007:**
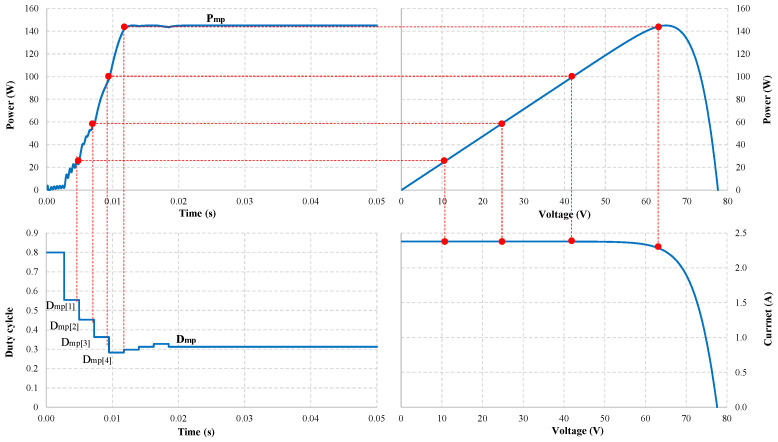
Location of MPPs (red dots) on the output waveform under uniform conditions.

**Figure 8 sensors-25-01908-f008:**
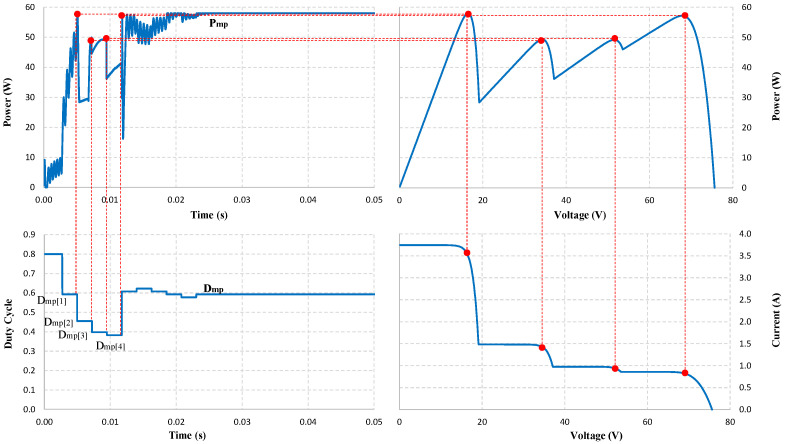
Location of MPPs (red dots) on the output waveform under PSC.

**Figure 9 sensors-25-01908-f009:**
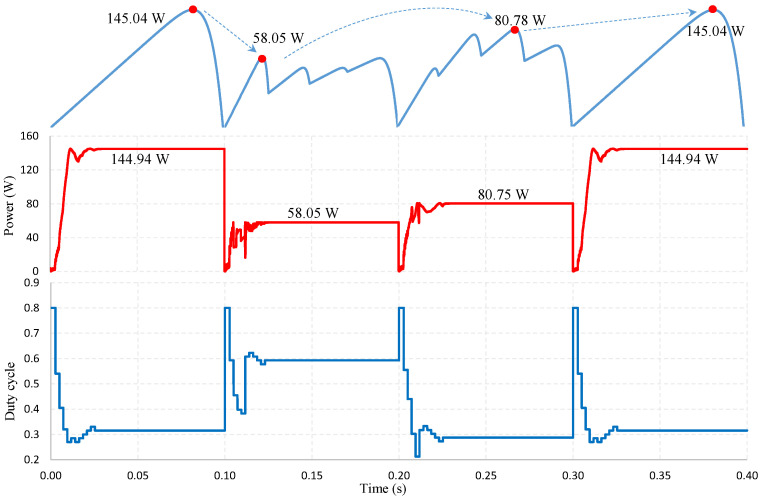
MPPT under continuously changing operating conditions.

**Figure 10 sensors-25-01908-f010:**
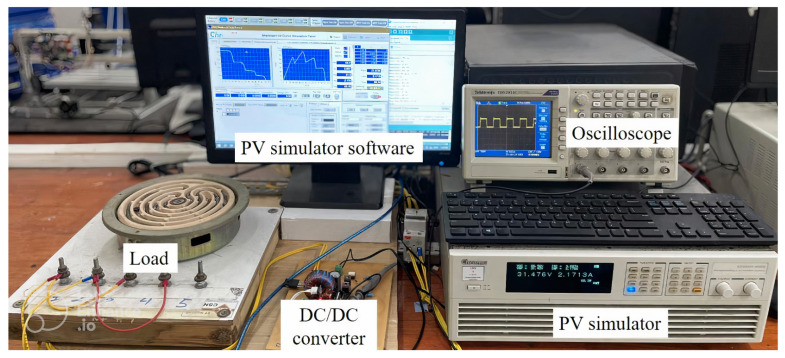
Experimental setup.

**Figure 11 sensors-25-01908-f011:**
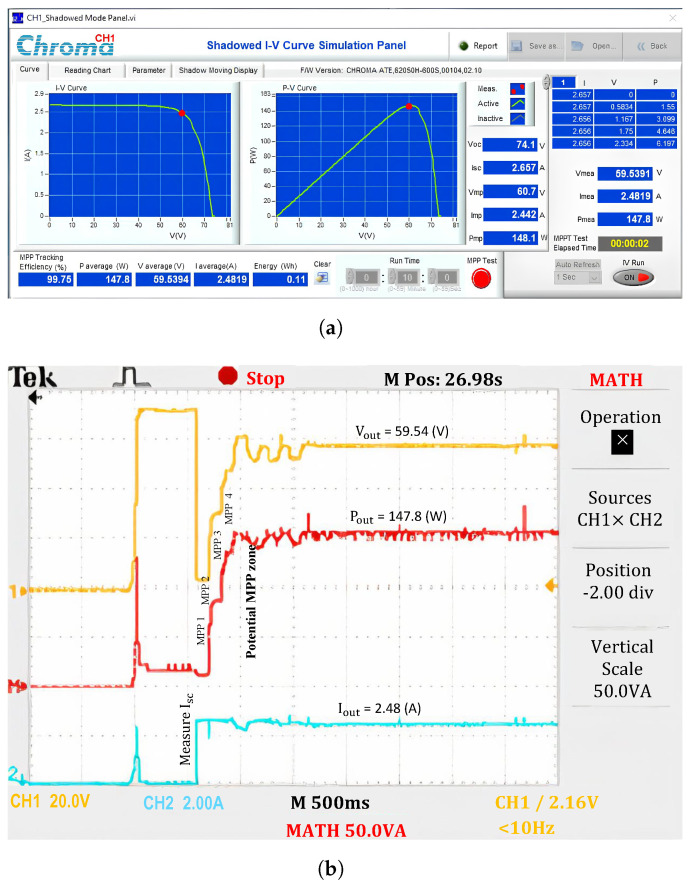
Experimental results with the proposed algorithm under uniform conditions. (**a**) MPP value, and (**b**) output waveforms.

**Figure 12 sensors-25-01908-f012:**
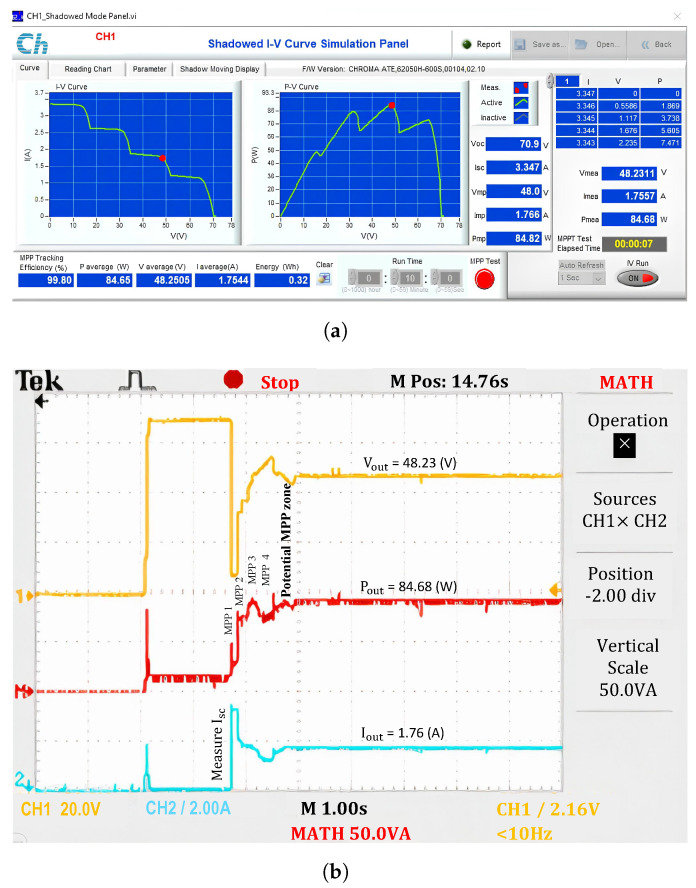
Experimental results with the proposed algorithm under PSCs. (**a**) MPP value, and (**b**) output waveforms.

**Figure 13 sensors-25-01908-f013:**
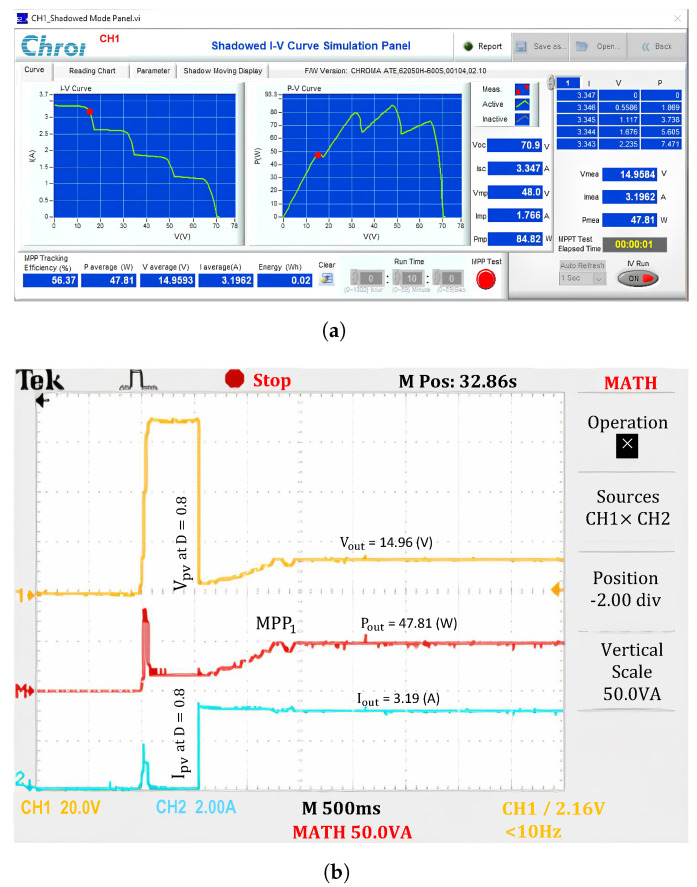
Experimental results with the P&O under PSCs. (**a**) MPP value, and (**b**) output waveforms.

**Figure 14 sensors-25-01908-f014:**
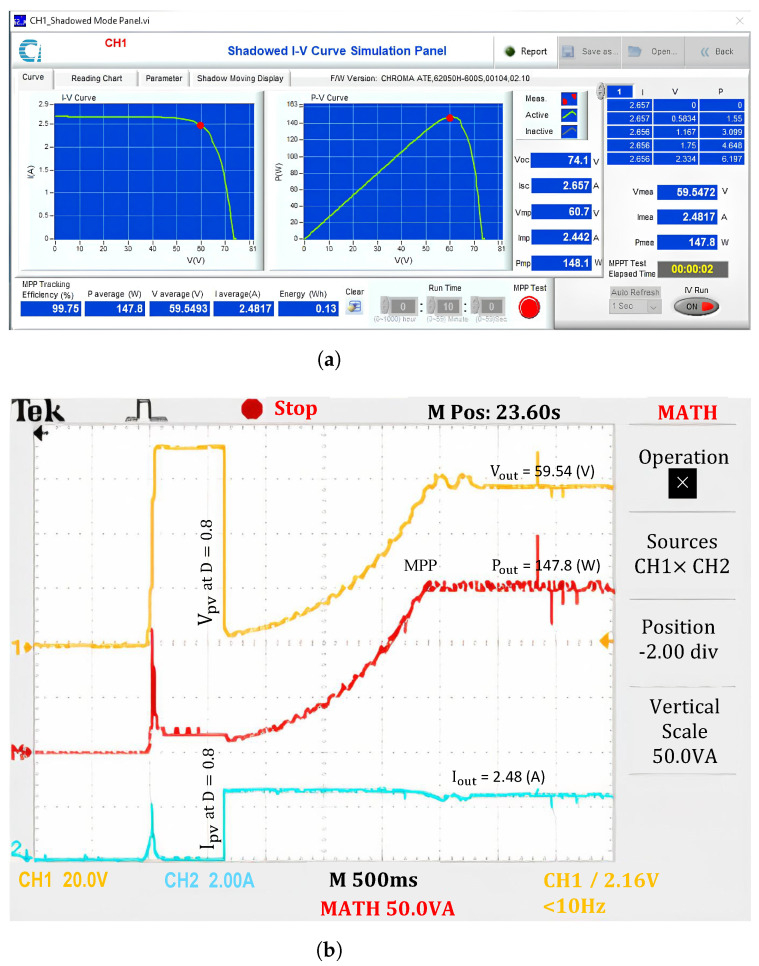
Experimental results with the P&O uniform conditions. (**a**) MPP value, and (**b**) output waveforms.

**Table 1 sensors-25-01908-t001:** Compare MPPT algorithms [1,2].

GMPPT Method	Speed	Efficiency	PSCs	Accuracy	Complexity	Stability	Cost
ANN-P&O	VH	-	Y	H	H	H	H
PSO-P&O/InC	VH	98.00	Y	H	M	M	M
GWO-ANFIS [14]	H	98.20	Y	M	M	H	H
AFO [8]	M	98.70	Y	M	H	M	H
PSO	H	-	Y	M	H	H	M
ABC	H	99.80	Y	M	H	H	H
ACO	H	-	Y	M	H	H	M
ANN [15]	M	99.81	Y	H	H	H	H
BA	H	99.90	Y	H	H	H	H
GWO [9]	M	99.57	Y	H	H	H	M
GA	M	-	Y	H	H	H	H
P&O	L	98.98	N	M	L	N	M
InC	L	99.94	N	M	M	Y	M
CV	L	100.00	N	L	L	N	L
CC [16]	L	99.88	N	L	L	N	L
Proposed method	VH	100.00	Y	H	L	H	L

VH: Very High; H: High; M: Medium; L: Low; Y: Yes; N: No.

**Table 2 sensors-25-01908-t002:** Cases study on the proposed algorithm.

No.	${\mathrm{PV}}_{1}$ (W/$m^{2}$)	${\mathrm{PV}}_{2}$ (W/$m^{2}$)	${\mathrm{PV}}_{3}$ (W/$m^{2}$)	${\mathrm{PV}}_{4}$ (W/$m^{2}$)	GMPP Location	$P_{\max}$ (W)
1	610	610	610	610	4	145.04
2	250	800	410	600	3	80.78
3	330	360	400	500	4	83.25
4	200	840	300	800	2	98.17
5	250	220	380	960	1	58.05

**Table 3 sensors-25-01908-t003:** Simulation results using the proposed algorithm.

No.	$D_{\mathrm{ref}}$	$D_{\mathrm{mp}}$	**|$D_{\mathrm{ref}}$ − $D_{\mathrm{mp}}$|**	$P_{\mathrm{ref}}$ (W)	$P_{\mathrm{mp}}$ (W)	ΔP%	MPPT Eff (%)	Time (ms)
1	0.2850	0.3150	0.030	144.73	144.94	0.14	99.93	18.93
2	0.3175	0.2875	0.030	79.07	80.75	2.08	99.96	19.91
3	0.1975	0.2275	0.030	77.43	83.24	6.98	99.99	21.45
4	0.5150	0.4850	0.030	91.74	98.16	6.54	99.99	21.14
5	0.6075	0.5925	0.015	57.07	58.05	1.69	100.00	25.71

**Table 4 sensors-25-01908-t004:** MPPT experimental performance.

Cases Study	1	2	3	4	5
Power (W)	147.8	84.68	82.61	97.81	57.47
MPPT efficiency (%)	99.75	99.80	99.23	99.63	99.00

**Table 5 sensors-25-01908-t005:** Compare MPPT methods.

No.	GMPPT Method	Efficiency (%)	Convergence Speed (s)	PSC Testing
1	Proposed Method	100.00	0.018	Y
2	GA - FOCV [5]	99.96	0.070	Y
3	InC - PSO [6]	99.07	0.0434	N
4	AFO [8]	98.70	0.880	Y
5	GWO [9]	99.57	72	Y
6	LS-C [11]	99.99	0.080	Y
7	CV [12]	99.80	0.150	Y
8	M-VSS-P&O [13]	99.37	0.016	N
9	CV [1]	100.00	-	N
10	GWO-ANFIS [14]	98.20	0.020	Y
11	CC [16]	99.88	0.700	N
12	PSO [25]	99.96	0.920	Y

## Data Availability

The data presented in this study is available on request from the corresponding author.

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
