# Peer review of "A Hybrid P&O and PV Characteristics Simulation Method for GMPPT in PV Systems Under Partial Shading Conditions"

_sensors, 2025, doi:10.3390/s25061908_

Round 1
Reviewer 1 Report
Comments and Suggestions for Authors
The article is interesting and of a high standard. It proposes a quite reasonable (fast, accurate, simple, and low-cost) solution to the global maximum power point (GMPPT) tracking problem based on partial shading. Critical remarksApplies to a paragraph “The literature [12] introduced an improved version of the Constant Voltage (CV) method, which uses a single parameter (tmax) to be automatically updated after each iteration. Although the solution can improve the convergence speed by up to 38.75% compared to the original version, it does not guarantee that each D value, D = 0.2, 0.4, 0.6, and 0.8, will identify a value in an extreme region”.
Unfortunately, the D parameter is not clearly defined here (it is not defined at all). Also, it is not defined (what it is, what is its importance) later in the article, but it is one of the most important parameters of your GMPPT method and algorithm.
Therefore, please describe clearly and precisely what the D parameter is.
The article also needs small linguistic corrections.
Comments on the Quality of English LanguageNA
Author Response
Comments 1: The article is interesting and of a high standard. It proposes a quite reasonable (fast, accurate, simple, and low-cost) solution to the global maximum power point (GMPPT) tracking problem based on partial shading. Critical remarks Applies to a paragraph “The literature [12] introduced an improved version of the Constant Voltage (CV) method, which uses a single parameter (tmax) to be automatically updated after each iteration. Although the solution can improve the convergence speed by up to 38.75% compared to the original version, it does not guarantee that each D value, D = 0.2, 0.4, 0.6, and 0.8, will identify a value in an extreme region”.
Unfortunately, the D parameter is not clearly defined here (it is not defined at all). Also, it is not defined (what it is, what is its importance) later in the article, but it is one of the most important parameters of your GMPPT method and algorithm.
Therefore, please describe clearly and precisely what the D parameter is.
The article also needs small linguistic corrections
Response 1: Thank you for pointing this out, which will help us present our research’s contributions and aims more clearly. We agree with this comment. Therefore, we have updated the manuscript by adding a detailed definition and explanation of the D parameter. Please see the details in the manuscript on page 4, paragraph 1 (from lines 120 to 135) or in the citation below.
“DC/DC converter
A PV system can maximize energy extraction by adjusting the load resistance (RL) to match the characteristic resistance at the MPP. However, directly controlling RL is challenging due to constantly changing operating conditions and the need to maintain stable power delivery to the load. Therefore, DC/DC converters are crucial in achieving this goal. These converters regulate the duty cycle (D) based on the ratio between the output voltage (Vout) and the input voltage (Vin) to adjust the equivalent resistance of the PV system, bringing it closer to the MPP. Each duty cycle value (0 < D < 1) corresponds to an operating point on the I-V curve. As operating conditions change, the MPP on the I-V curve shifts, requiring an adjustment in D. To respond quickly to rapid environmental changes, the MPPT controller employs automatic search algorithms to dynamically adjust the pulse width, allowing the system to quickly track the MPP. Among various types of DC/DC converters, the Buck-Boost converter is particularly advantageous when the PV system's output voltage varies over a wide range. It can operate in both buck and boost modes, allowing it to track GMPP throughout the PV curve [12]. The following equation gives the duty cycle of the Buck-Boost converter.
Thank you so much! We appreciate and welcome your valuable feedback. We have made efforts to improve the expression to convey information to readers as clearly as possible. Therefore, the entire manuscript has been revised and adjusted based on your recommendations.

Reviewer 2 Report
Comments and Suggestions for Authors
This study reported a GMPPT solution for a series-connected PV system operating under PSC and continuously changing operating conditions, which has high simulation efficiency and experimental efficiency, showing promising potential applications in PV systems operating under different conditions. Revisions are required before publication consideration.
- What does D values mean in the paper? Please address the importance of D value on the simulation. Why could a random D value be selected to do the calculation?
- On page 7, line 201, what does M2 present?
- How does solar radiation affect the calculation results?
- Why could the shortcomings in previous studies be overcome by referring to 0.4Voc position to test PSC in this study?
- In this study, the authors mentioned that the simulation or calculation was conducted under homogeneous conditions, are there any chances that the authors would consider the heterogenous conditions while doing the simulation and calculations? How would the heterogeneous conditions affect the results?
Author Response
Comments 1: What does D values mean in the paper? Please address the importance of D value on the simulation. Why could a random D value be selected to do the calculation?
Response 1: Thank you very much for your comment, which will help us present our research’s contributions and aims more clearly. We agree with this comment. Therefore, we have updated the manuscript by adding a detailed definition and explanation of the D parameter. Please see the details in the manuscript on page 4, paragraph 1 (from lines 120 to 135), and page 6, paragraph 5 (from lines 182 to 191), or in the citation below.
“DC/DC converter
A PV system can maximize energy extraction by adjusting the load resistance (RL) to match the characteristic resistance at the MPP. However, directly controlling RL is challenging due to constantly changing operating conditions and the need to maintain stable power delivery to the load. Therefore, DC/DC converters are crucial in achieving this goal. These converters regulate the duty cycle (D) based on the ratio between the output voltage (Vout) and the input voltage (Vin) to adjust the equivalent resistance of the PV system, bringing it closer to the MPP. Each duty cycle value (0 < D < 1) corresponds to an operating point on the I-V curve. As operating conditions change, the MPP on the I-V curve shifts, requiring an adjustment in D. To respond quickly to rapid environmental changes, the MPPT controller employs automatic search algorithms to dynamically adjust the pulse width, allowing the system to quickly track the MPP. Among various types of DC/DC converters, the Buck-Boost converter is particularly advantageous when the PV system's output voltage varies over a wide range. It can operate in both buck and boost modes, allowing it to track GMPP throughout the PV curve [12]. The following equation gives the duty cycle of the Buck-Boost converter.
However, this problem is addressed using measurement results obtained from a random D value within the linear region of the I-V curve. This method enables the proposed algorithm to achieve two calculation objectives simultaneously. First, the current in the linear region is used to determine Isc based on the relationship Isc = Ipv. This result can be further explained by the fact that, within the voltage range of less than 0.4Voc, the I-V curve exhibits linear behavior [22]. To measure the parameters in this linear region, the D value of the Buck-Boost converter must be greater than 0.69 [23]. The newly measured Isc value is then used to calculate the Voc of the PV panel according to (4). Then, Rmp, the load resistance (RL) are determined using Equations (6) and (13), respectively. Finally, the duty cycle at the MPP (Dmp) is computed according to Equation (12).
Comments 2: On page 7, line 201, what does M2 present?
Response 2: Thank you for pointing this out. We agree with this comment. Therefore, we have added a detailed definition and explanation of the M2 parameter in the manuscript on pages 7 and 8, (from lines 224 to 231). Please also see the details of the adjustment in the excerpt below.
“Assuming the PV system operates under uniform conditions, the value Voc[2] = 2Voc[1], so at the MPP of the second step region (M2), the resistance value (Rmp[2]) is estimated according to (14). In this case, the line representing Rmp[2] intersects the I-V curve at point A, which coincides with M2. However, suppose the PV system operates under inhomogeneous conditions. In that case, the MPP of the second step region will shift to M2’ or M2”, depending on the shading level. So, the intersection point between the line representing Rmp[2] and the I-V curve will be A’ or A", which does not coincide with M2′ or M2” (Figure 4)”
Comments 3: How does solar radiation affect the calculation results?
Response 3: Thank you very much for your comment on this important aspect related to MPPT algorithms. Your suggestion has given us the opportunity to present our research results more clearly. We have added the missing part to the manuscript on pages 12 and 13, paragraph 1 (from lines 361 to 378). Please see the details in the manuscript or in the citation below.
“When the irradiance changes, the Isc and Voc parameters of the PV system fluctuate but have little impact on determining the first MPP position. This property results from Isc being directly proportional to solar irradiance. If Isc is accurately determined, Voc will be precisely calculated using equation (3). As analyzed earlier, with the initial duty cycle D selected in this study, the measurement of Isc is consistently ensured to be within the linear region. Consequently, the error in calculating Voc and the MPP position is negligible. The I-V characteristics in Figure 6 also indicate that the voltage Voc[1] remains relatively stable under all test conditions, and the voltage at the first MPP position, Vmp[1], is nearly constant. This result facilitates the accurate determination of Dmp[1] and serves as a reference value for implementing a partial shading detection algorithm on the I-V curve.
However, varying irradiance conditions influence the speed of shading detection in the PV system. This phenomenon can be explained using Figure 4: Under a small decrease in irradiance, a slight adjustment of the resistance value is enough to move from the intersection point A′ to position M2′. In contrast, under a significant irradiance decrease, moving from A′′ to M2′′ requires multiple resistance updates. Nonetheless, resistance adjustments are performed independently of the duty cycle search loops. Therefore, the proposed method ensures output waveform stability, as the adjustments to D are only performed around GMPP. The following simulation and experimental results will validate this approach.”
Comments 4: Why could the shortcomings in previous studies be overcome by referring to 0.4Voc position to test PSC in this study?
Response 4: Thank you for your insightful question. It helps us highlight the contributions of our research more clearly. We have added an explanation to the manuscript on pages 8 and 9, paragraph 4 (from lines 273 to 290). Please see the details in the manuscript or in the citation below.
“This study overcomes the shortcomings of previous research by using the 0.4Voc position for PSC testing. The justification for using the proposed 0.4Voc point in PSC checking is as follows: First, when irradiance decreases, the line containing Rmp[2] can intersect the I-V characteristic at either A' or A", depending on the percentage of shading (Figure 4). If the system operates uniformly or with light partial shading, the measured voltages at A or A′ are greater than Voc[1]+0.4Voc[1]. Consequently, the current values at these points are selected as the short-circuit currents in the second region (Isc[2]). These currents are also higher than those at the MPP in the second region (at ?2′ or M2′′), ensuring that the MPP is located within this region without shifting into the third region. Second, within the 0.4Voc limit, the I-V characteristic remains linear, making the determination of Isc in this region more accurate. However, under significantly reduced irradiance (represented by the red characteristic line), the operating point A" may retreat into the nonlinear region (from M1 to Voc[1]). If the measured values in this region are used to determine the MPP, they remain in Region 1. In these cases, shifting the operating point to region 2 is necessary by increasing the resistance value Rmp[2]. Therefore, the 0.4Voc threshold is an optimal limit for updating the operating region, ensuring no step region is overlooked. To transition to the second step region, the Rmp[2] value should be increased according to equation (14), and the Dmp[2] value should be recalculated using equation (11) until VA" > 1.4Voc[1].”
Comments 5: In this study, the authors mentioned that the simulation or calculation was conducted under homogeneous conditions, are there any chances that the authors would consider the heterogenous conditions while doing the simulation and calculations? How would the heterogeneous conditions affect the results?
Response 5: Thank you for your valuable feedback. This issue indicates that certain aspects of our manuscript were not clearly presented. We appreciate the opportunity to provide a clearer and more comprehensive version. We have revised the manuscript and included the missing section on pages 7 and 8, (from lines 224 to 231). Please also see the details of the adjustment in the excerpt below.
“Assuming the PV system operates under uniform conditions, the value of Voc[2] = 2Voc[1], so at the MPP of the second step region (Mâ‚‚), the resistance value is estimated according to equation (13). In this case, the line representing Rmp[2] intersects the I-V curve at point A, which coincides with Mâ‚‚.
However, suppose the PV system operates under non-uniform conditions. In that case, the MPP of the second step region will shift to Mâ‚‚' or Mâ‚‚", depending on the shading level. So the intersection point between the line representing Rmp[2] and the I-V curve will be A' or A", which does not coincide with Mâ‚‚' or Mâ‚‚" (Figure 4).”
5. Additional clarifications
In this study, the simulation and experimental scenarios were primarily conducted under heterogeneous conditions. Among the five proposed cases (Table 2, page 12), only Case 1 operates under homogeneous conditions. This task is also illustrated in Figures 6, 8, and 9, (on pages 12, 13 and 15) with different numbers of LMPPs and GMPP locations corresponding to various voltage regions across the entire I-V curve. These scenarios demonstrate the possibility of extracting GMPPs at the lowest voltage (first peak) and highest voltage (last peak) regions, where the values of D need to approach extreme points of 0 or 1 to ensure efficiency.

Reviewer 3 Report
Comments and Suggestions for Authors
The topic of the paper is interesting, and the manuscript is well-written. However, I believe there is still significant room for improvement. Please consider the following comments:
- The title of the paper is too general and sounds more like a book chapter. In my opinion, including the algorithm's name in the title is not particularly meaningful. I suggest proposing an alternative title.
- Abbreviations defined in the abstract should be redefined in the introduction to improve readability.
- I appreciate the simulation analysis and experimental results; however, several important aspects are missing:
- The MPPT algorithm is executed on a control board that regulates the duty cycle of the DC/DC converter. However, I could not find any details regarding the control of the converter, which is crucial for a fair comparison of algorithms. A proper comparison requires a clear description of the execution steps of each algorithm, along with all relevant parameters.
- On what basis do you conclude that the proposed algorithm has low complexity and low implementation costs? Why would it have lower costs compared to a simple P&O algorithm, for example?
- Please provide detailed information on the equipment used, such as the specific model of the Chroma device.
- It would be interesting to see how this algorithm performs at higher voltages and currents, where measurement noise is more pronounced. A key question is how robust the algorithm is to noise in the measured signal.
- Could this algorithm be applied to other converter topologies, such as inverters?
English is fine.
Author Response
Comments 1: The title of the paper is too general and sounds more like a book chapter. In my opinion, including the algorithm's name in the title is not particularly meaningful. I suggest proposing an alternative title.
Response 1: Thank you for your valuable suggestion. We acknowledge that the original title may be too general and could be improved for better clarity and specificity. Based on your feedback, we have revised the title to better reflect the focus of our study while maintaining its relevance. The new proposed title is: 'A new hybrid algorithm combining P&O and the PV characteristics simulation method for GMPPT in a PV system under partial shading conditions'. We hope this revised title addresses your concern and provides a clearer information of our work.
Comments 2: Abbreviations defined in the abstract should be redefined in the introduction to improve readability.
Response 2: Thank you for your suggestion. We acknowledge that redefining abbreviations in the introduction can improve readability. Based on your feedback, we have reintroduced the key abbreviations in the introduction to ensure clarity for readers. Please see the details in the manuscript on pages 1 to 3 or some additional definitions cited below.
“partial shading conditions (PSC), local maximum power point (LMPP), global maximum power point tracking (GMPPT), maximum power point tracking (MPPT), duty cycle (D)”.
Comments 3: I appreciate the simulation analysis and experimental results; however, several important aspects are missing:
The MPPT algorithm is executed on a control board that regulates the duty cycle of the DC/DC converter. However, I could not find any details regarding the control of the converter, which is crucial for a fair comparison of algorithms. A proper comparison requires a clear description of the execution steps of each algorithm, along with all relevant parameters.
On what basis do you conclude that the proposed algorithm has low complexity and low implementation costs? Why would it have lower costs compared to a simple P&O algorithm, for example?
Please provide detailed information on the equipment used, such as the specific model of the Chroma device.
It would be interesting to see how this algorithm performs at higher voltages and currents, where measurement noise is more pronounced. A key question is how robust the algorithm is to noise in the measured signal.
Could this algorithm be applied to other converter topologies, such as inverters?
Response 3: Thank you for your valuable comments and for appreciating our simulation analysis and experimental results. We appreciate your valuable input and agree to supplement the shortcomings related to this issue on page 4, paragraph 1 (from lines 120 to 135)
“DC/DC converter
A PV system can maximize energy extraction by adjusting the load resistance (RL) to match the characteristic resistance at the MPP. However, directly controlling RL is challenging due to continuously changing operating conditions and the need to stabilize power delivery to the load. Therefore, DC/DC converters are crucial in achieving this goal. These converters regulate the duty cycle (D) based on the ratio between the output voltage (Vout) and the input voltage (Vin) to adjust the equivalent resistance of the PV system, bringing it closer to the MPP. Each duty cycle value (0 < D < 1) corresponds to an operating point on the I-V curve. As operating conditions change, the MPP on the I-V curve shifts, requiring an adjustment in D. To respond promptly to rapid environmental changes, the MPPT controller employs automatic search algorithms to dynamically adjust the pulse width, ensuring the system quickly tracks the MPP. Among various types of DC/DC converters, the Buck-Boost converter is particularly advantageous when the PV system's output voltage varies over a wide range. It can operate in buck and boost modes, allowing it to track the GMPP across the entire P-V curve [12]. The following equation gives the duty cycle of the Buck-Boost converter.
And on page 15, paragraph 2 (from lines 438 to 441) and page 16, paragraph 2 (from lines 459 to 466). Please also see the details of the adjustment in the excerpt below.
The proposed algorithm is tested using a PV Chroma 62050H-600S simulator for the cases in Table 2. This study employs a buck-boost converter to control the PV system operating at the GMPP. The main parameters of the DC/DC converter include an input capacitance (Cin=100 µF), an output capacitance (Cout = 47 µF), an inductance (L = 0.4 mH), and a load resistance of RL = 20Ω.
The proposed solution follows the same operating principle as the conventional P&O algorithm, which has been demonstrated to be less complex than other algorithms [14, 15]. Additionally, it reduces costs compared to the unimproved version, as it does not require separate voltage and current sensors. Instead, these parameters are directly computed from a single initial D under specific operating conditions. As a result, the proposed method not only improves performance and convergence speed under PSC but is also simple, easy to implement, and highly accurate.”
5. Additional clarifications
GMPP is determined as the maximum value among all LMPPs. Therefore, any measurement error may affect the GMPP value. However, such errors only impact potential LMPPs with similar power levels (e.g., case 5 in Figure 8). In these situations, misidentification of the GMPP may occur, but the output maximum power deviation is negligible. For instance, measurement noise may cause the algorithm to misidentify the GMPP location (Figure 8). It may select the fourth peak instead of the first peak. However, the GMPPT performance is not significantly reduced since the power values at these two peaks are nearly identical. In practical scenarios, if a measurement error occurs, any algorithm may fail to correctly identify the GMPP when the power levels of LMPPs are close. Nevertheless, no errors were observed when testing the proposed algorithm on the Chroma 62050H-600S solar simulator under the conditions considered in this study (Figure 11b).
This study uses the MPPT algorithm to regulate the DC/DC converter for optimal PV system power. Although inverters with built-in MPPT aim to optimize power extraction from the source, their control method differs slightly. If MPPT is applied at the DC/DC stage before supplying power to the inverter, the proposed MPPT algorithm can be applied with appropriate adjustments. Since the power injected into the grid by the inverter matches the maximum power extracted by the DC/DC converter, the DC bus voltage remains constant. Duty cycle D is computed based on the output voltage ratio of the PV system's measured voltage rather than on resistance. Conversely, applying this algorithm directly to an inverter requires considering the system's specific structure, which could be explored in future research. Therefore, we would like to remain the conclusions of the previous manuscript.

Round 2
Reviewer 2 Report
Comments and Suggestions for Authors
The authors have fully addressed the reviewer's comments. It could be accepted in the present form.
Author Response
Comments 1: The authors have fully addressed the reviewer's comments. It could be accepted in the present form.
Response 1: Thank you so much!

Reviewer 3 Report
Comments and Suggestions for Authors
Thank you for your responses and valuable comments. However, some aspects of the experimental results are still unclear to me.
Could you specify which microprocessor you used to control the DC-DC converter? How did you implement the variation of the duty cycle factor, and what switching frequencies did you use?
The paper lacks sufficient experimental results—there is only one figure, which I believe is inadequate. Please include more experimental results, for example, by capturing and presenting waveforms using an oscilloscope. It would be useful to observe the dynamic response of maximum power point tracking (MPPT), specifically how the voltage and current change and how quickly the algorithm reaches the MPP. Additionally, this response should be compared with the response of the conventional P&O algorithm or another algorithm. The superiority of your algorithm must be demonstrated through experimental results, based on response analysis rather than unverified tables.
If necessary, remove some simulation results and add more experimental ones, as they are crucial for validation. It is also essential to explain how the DC-DC converter algorithm was implemented, as this will provide insight into its complexity.
Author Response
Comments 1: Thank you for your responses and valuable comments. However, some aspects of the experimental results are still unclear to me.
Could you specify which microprocessor you used to control the DC-DC converter? How did you implement the variation of the duty cycle factor, and what switching frequencies did you use?
The paper lacks sufficient experimental results—there is only one figure, which I believe is inadequate. Please include more experimental results, for example, by capturing and presenting waveforms using an oscilloscope. It would be useful to observe the dynamic response of maximum power point tracking (MPPT), specifically how the voltage and current change and how quickly the algorithm reaches the MPP. Additionally, this response should be compared with the response of the conventional P&O algorithm or another algorithm. The superiority of your algorithm must be demonstrated through experimental results, based on response analysis rather than unverified tables.
If necessary, remove some simulation results and add more experimental ones, as they are crucial for validation. It is also essential to explain how the DC-DC converter algorithm was implemented, as this will provide insight into its complexity.
Response 1: Thank you for your insightful question. Your suggestion helps us highlight the contributions of our research more clearly. Based on your feedback, we have added the missing part to the manuscript on pages 17 to 20 (from lines 442 to 516). Please see the details in the manuscript or in the citation below.
The main parameters of the DC/DC converter include an input capacitance (Cin = 100 µF), an output capacitance (Cout = 47 µF), an inductance (L = 0.4 mH), and a load resistance of RL = 20 Ω, the switching frequency is 50 kHz. In this study, an Arduino Nano ATmega328P controls the DC/DC converter, providing an optimal D value for the PV system operating at the GMPP. The experimental setup, as shown in Figure 10, is used to analyze the PV system’s output waveforms under different conditions. A summary of the output power and experimental performance is presented in Table 4, showing an average GMPPT efficiency of approximately 99.48%, with a maximum extraction of 99.80%. The specific experimental scenarios are as follows.
Figure 11. Experimental results with the proposed algorithm under uniform conditions. a. MPP
value, and b. output waveforms (Please see details in attached file)
When operating under uniform conditions (No.1), the output power is approximately 99.75% the maximum power of the PV system. Figure 11 shows the output waveform obtained under homogeneous conditions. The results indicate that the short circuit current value of the PV system, determined from the initial position D = 0.8, is also the short circuit current in the first step region (Isc[1]). Then, the LMPP values in the regions are determined and updated sequentially. The current waveform shows that the Isc[i] values in different regions do not differ significantly. However, the current in zone 4 is slightly lower than in the other regions because the measurement point falls within the transition region between the linear and nonlinear segments of the I-V curve. Meanwhile, the voltage and power waveforms continue to increase steadily because the solution searches from low-voltage to high-voltage regions, and only one extremum exists on the P-V curve. The potential GMPP limit belongs to Zone 4, and the proposed solution performs a maximum power value check there based on the P&O method. Several adjustment steps are required to ensure the maximum possible output power extraction. Since there is no significant current drop between regions, the calculation points do not need to update the increment ∆R to adjust the operating region.
Figure 12. Experimental results with the proposed algorithm under PSC. a. MPP value, and b. output
waveforms (Please see details in attached file)
When operating under PSC (No.2), the principle of determining Isc and the MPP in the first step region remains unchanged. However, the current tends to decrease in the subsequent regions (Figure 12). The current in Zone 2 is lower than in Zone 1, but the maximum power still increases due to the increasing voltage level. The same phenomenon occurs in Region 3. However, when transitioning from Region 2 to Region 3, the solution must adjust the resistor to ensure that the two measured maximum values do not belong to the same step region (i.e., they do not have the same current value). This operating mechanism repeats when switching the search region from 3 to 4. At the end of the PSC checking, the MPP in Region 3 is the highest compared to the other regions. Therefore, the algorithm confines the potential GMPP to this region to optimize the search for the GMPP value. The current waveform in Figure 12b shows that after scanning for PSC across the entire I-V curve, the current reverses from the lowest region, Region 4, back to Region 3. Meanwhile, the voltage in Region 3 is lower than in Region 4, causing the output voltage waveform to drop suddenly. As a result, the power reaches its maximum efficiency of approximately 99.80%.
Figure 13. Experimental results with the P&O under PSC. a. MPP value, and b. output waveforms (Please see details in attached file)
To compare the effectiveness of the proposed solution with the traditional version, both versions use the same implementation principle of the P&O algorithm. They are initialized with a starting value of D = 0.8 and a step size of ∆D = 0.015. The key difference is that the traditional P&O algorithm searches directly from the starting position until it finds the nearest extreme value. Meanwhile, the proposed solution performs independent calculations to determine the LMPPs and identify the potential GMPP limit in the first stage. After narrowing the search area, the P&O algorithm is then deployed to determine the convergence point. The experimentally obtained output waveforms demonstrate that the proposed solution is more effective than the traditional P&O method in the following aspects: First, it has a superior ability to avoid LMPP traps by simulating I-V characteristics under different operating conditions. The potential GMPP region is identified among the LMPPs to ensure that no extreme points on the I-V curve are overlooked. While, the traditional P&O method always identifies the first extremum in the low voltage region when starting from the initial value, D = 0.8, (Figure 13). This principle indicates that the traditional P&O method can correctly identify only two cases: No.1 (where there is a single extremum) and No.5 (where the extremum belongs to Region 1). Therefore, the experimental results for Case 2 are trapped in LMPP, resulting in an efficiency only slightly above 56.37%.
Figure 14. Experimental results with the P&O uniform conditions. a. MPP value, and b. output
waveforms (Please see details in attached file)
Second, the output signal of the proposed method is more stable than that of the traditional P&O method. In the old version, the output waveform is affected by continuous fluctuations due to changes in step size ∆D. Therefore, the convergence speed and performance of the traditional P&O method depend on this parameter. A small ∆D will improve the performance of the PV system, but numerous adjustments are required, causing continuous fluctuations in the output waveform and a slow convergence speed. In contrast, if ∆D is large, fewer iterations are needed, resulting in faster convergence; however, efficiency decreases, and the output signal at the operating point becomes unstable. This drawback can be observed in the output waveform shown in Figure 14. Although it achieves MPPT efficiency equivalent to that of the proposed solution due to the presence of only one extremum in the I-V characteristic, its output waveform converges 1 second slower. The proposed solution mitigates this drawback by using a significant adjustment of ∆R to quickly change the search area. The value of D remains stable while updating the new position based on R, minimizing the number of iterations. Therefore, the proposed solution can utilize a small ∆D value to enhance GMPPT performance and stabilize the output signal without compromising the convergence speed. The output power waveform in Figure 11 shows this feature. Although there is only one extremum, the solution introduces four resting periods to reposition the search area, whereas the traditional P&O method continuously loops to reach the MPP.
